# Revitalize Traditional Agriculture: Chinese Consumer Perception and Preference of "Modern" Organic and Sustainable Traditional Rice Products

**Erpeng Wang**

School of Economics and Management, Nanjing Tech University, Nanjing 211816, China; 5637@njtech.edu.cn

**Abstract:** Many smallholder farmers in developing countries have used sustainable traditional agricultural techniques to ensure food safety and sustainability over the centuries. However, the value of sustainable traditional agricultural products, especially as an inexpensive substitution for "modern" organic products in developing countries, is rarely studied. Using the contingent valuation method, we compared Chinese consumers' perceptions of and preferences for sustainable traditional agricultural products and "modern" organic products. Our results show that sustainable traditional agriculture can signal environment-friendly practices and food safety, and consumer willingness to pay for sustainable traditional agricultural products is higher than those of "modern" organic products. Considering the high demand for sustainable traditional agricultural products, revitalizing sustainable traditional agriculture may be a good way to balance sustainability and feasibility in developing countries.

**Keywords:** "modern" organic agricultural products; sustainable traditional agricultural products; payment card; willingness to pay

## 1. Introduction

The general public has been greatly interested in sustainable products because of rising concerns regarding food safety and environmental issues. In response to the changes in market demand, the food industry has offered more organic agricultural products than before. The past few years have witnessed the rapid expansion of the organic market worldwide. Between 2000 and 2018, the global sale of organic products grew from 18 billion dollars to 95 billion dollars [1]. In many countries, growers and agro-food companies are undergoing a rapid transformation into organic farming, attempting to capitalize on the fast-growing demand [2]. In 2019, the global land for organic farming reached 72.3 million hectares, more than twice that of 2007, at 32.2 million hectares [3,4]. Since 2008, the proportion of organic food in total food sales in the United States has increased yearly. In 2017, organic food sales accounted for 5.5% of the total in the United States [1]. Among all the agricultural production segments, organic farming is the part that grows fastest.

These "modern" organic products are certified by third-party agents and sold with their logo to assure quality and promote commerce. Most current literature focuses on consumer attitudes toward "modern" organic products [5–8]. These studies show that consumers are willing to pay more for organic food than conventional food. Most studies report that the key driving factor for buying organic meat is the belief that it has higher safety, health, and welfare standards and that it is more environment-friendly [9,10]. The information on the production method also affects consumers' willingness to pay (WTP) for organic meat [11]. However, according to a meta-analysis of 96 observations, the WTP values of organic food are very different: the average premium is 36%, the highest value is 509.2%, and the lowest value is only 2.3% [12]. Some people even do not prefer "modern" organic food due to perceived inferior quality regarding taste, safety, or appearance [13]. A review of selected research on consumer perceptions and understanding of organic food

further suggests that although consumers are generally organically aware, they do not have a consistent understanding of "organic" [14].

The value of agro-ecological features of sustainable traditional agriculture in developing countries is often neglected compared to the development of "modern" organic products. As a sort of farming practice, sustainable traditional agriculture takes advantage of the techniques evolving over hundreds of years to ensure yield in a specific area or region; they are often more sustainable and less polluting than modern conventional agricultural methods. Table 1 compares the characteristics of modern conventional agricultural products, sustainable traditional agricultural products, and "modern" organic agricultural products in terms of their production methods, environmental impact, certification and cost. A study by [15] shows that organic certification is not a prerequisite if consumers can obtain perceived high quality directly. People also became increasingly interested in the agro-ecological features of sustainable traditional agriculture. For instance, nine out of ten consumers in Belgium like free-range products and are willing to pay premiums between 43% and 93%, the highest among all the products examined in the study [15]. As is reported, Italian consumers would willingly pay a premium of €11.64/kg for beef and €10.25/kg for chicken if they are free-range products [16]. In addition, consumers in Europe would make an extra payment of 5% for the outdoor-produced pork and 20% for those labeled 'raised outside' [17].

**Table 1.** Comparison of different types of agricultural products.

| Category | Production Methods | Environmental Impact | Certification and Cost |
| --- | --- | --- | --- |
| Modern conventional agricultural products | Applying technologies and chemicals to increase production | Causing soil deterioration, over-fertilization, reduction in the diversity of cultivated crops, etc. | None; low cost but high externalities |
| Sustainable traditional agricultural products | Using traditional techniques evolving over hundreds of years to balance yield and sustainability | Being more sustainable and less polluting than modern agricultural methods | None or informal; low cost |
| "Modern" organic agricultural products | Following organic standards and certification systems | Being more sustainable and less polluting than modern agricultural methods | Formal and regulated; high cost |

Sustainable traditional agricultural products seem to be more popular than "modern" organic products in the Chinese food market. Research shows that there is a general perception among many Chinese consumers that the rice produced from the rice–fish system, a sustainable traditional agricultural production system, is environment-friendly and of high quality. Consumers would buy this type of rice with a premium of 41% [18]. Jingdong, one of China's top two B2C online retailers, as well as the major competitor to Tmall operated by Alibaba, has sold more than 3,650,000 packages of rice–crab co-culture system, but only about 110,000 packages of organic rice. Sustainable traditional farming is popular among numerous small-scale and family-managed farms in China and other developing countries such as Argentina, Thailand etc. They grow their products using techniques that improve nutrient flows and make better use of local resources such as local seeds and sustainable traditional knowledge (green manuring, fish–rice systems), which is consistent with the principles of organic farming. In both developed and developing countries, many farms do not certify their products as organic as the cost of certification [19]. For instance, in the United States, the number of certified organic farms has increased from 14,217 in 2016 to 16,585 in 2019 [20]. Small farms were more likely to decertify organic certification than large-sized farms [21]. However, food marketing experts often classify food as "modern" organic or not [15], which neglects the value of sustainable traditional farming, particularly those in developing countries regarding their contribution to sustainability and the ecosystem. Due to the increasing demand for

sustainable and environmentally friendly agricultural products [18,22], identifying and comparing consumer preferences for sustainable traditional agricultural products and "modern" organic products would provide important information to identify the market space for revitalizing sustainable traditional agriculture.

The raising question is whether consumers are willing to a premium for sustainable traditional agricultural products, and how about comparing to "modern" organic products. This study aims to estimate and compare Chinese consumers' perceptions of sustainable traditional agricultural and "modern" organic products and their attitudes towards them. This study estimates consumer preference for certified "modern" organic rice and two other kinds of rice from sustainable traditional farming (green manuring and fish–rice systems) with contingent valuation methods. Although numerous studies about consumer preference for organic products exist, comparisons between consumer preference for sustainable traditional agricultural products and that for "modern" organic products remain limited. To our knowledge, no studies compare consumers' preferences for sustainable traditional agricultural products and "modern" organic products in developing countries. The contribution of this research is to estimate and compare Chinese consumers' preferences of sustainable traditional agricultural and "modern" organic products, providing new insight into approaches for revitalizing sustainable traditional agriculture in developing countries. Result would help explore more feasible ways for smallholder farms in developing countries to achieve economic sustainability while at the same time contributing to the sustainability of the environment and ecosystem.

We organized the remainder of the paper as such. In the following section, the "organic" features of sustainable traditional agriculture were explained. Next, we put forward to research method, including theoretical model, WTP elicitation method and econometric model. Then, empirical results are reported with discussion. The last section is conclusion and implications.

## 2. The "Organic" Features of Sustainable Traditional Agriculture

In some developing countries, sustainable traditional agriculture is the most common players in the organic movement. Unlike the modern agricultural mode of applying technologies and chemicals to increase production [23], sustainable traditional agriculture, as a sort of farming, takes advantage of the techniques evolving over hundreds of years to ensure yield in a specific area or region year after year. Sustainable traditional agricultural techniques are usually adopted by smallholder farmers, especially in developing countries. They are often more sustainable and less polluting than modern agricultural methods [18,19]. The "organic" features of sustainable traditional agriculture provide a great potential opportunity to revitalize sustainable traditional agriculture.

Similar to organic production, most sustainable traditional farming practices seldom use synthetic chemical pesticides or fertilizers. They apply the techniques that improve nutrient flows and make better use of local resources such as local seeds and sustainable traditional knowledge. These different farming systems could be regarded as "organic", and they meet the needs of households and local markets with transparent information. For instance, in Argentina, although a large proportion of the commodities and cash crops aimed for export are produced with high-yield, capital-intensive practices, traditional agriculture that barely uses external inputs and modern technologies still prevails [19].

Beginning in the mid-20th, many countries' farming policies started incentivizing agriculture to adopt a "getting big or getting out" approach. This approach has changed the entire food system, driving traditional farming off the land in droves. However, the model agriculture that evolved mainly from the green revolution has resulted in many problems, including soil deterioration, over-fertilization, reduction in the diversity of cultivated crops, etc. With increasing concern over the side effect of modern agriculture, the potential market value of sustainable traditional farming is still large, and the diversification of agriculture should be treasured. The "organic" features of sustainable traditional agriculture are

gradually recognized by the market, especially when the e-commerce system is able to offer production information directly to consumers.

## 3. Research Method

### 3.1. Theoretical Model

We follow Lancaster's [24] consumer demand theory that the utility of a product is determined by the bundle of attributes rather than the product as a whole. Lancaster's consumer demand theory can be extended to random utility theory as a theoretical framework to analyze consumer choice [25]. Following previous studies about sustainable food [26] and green food [27], we assume there are four types of rice in the market: conventional rice, "modern" organic rice, rice from the fish–rice system, and rice fertilized by green manure. "Modern" organic rice is certified and labeled by organic labels, and rice from the fish–rice system or fertilized by green manure is claimed to be produced using sustainable traditional production technologies. We define consumer utility function over $E_{ij}$ rice as:

$$U(E_{ij}) = \alpha_0 \cdot P_{ij} + \alpha_{ij} \cdot E_{ij} + \beta_{ij} \cdot X_i \tag{1}$$

where $P_{ij}$ is the price of the rice $j$, $E_{ij}$ indicates the types of rice $j$, and $X_i$ is a vector of the demographic and perception variables, and $\beta_{ij}$ is the coefficient. Consumers' WTP for organic or traditional rice can be defined as the price premium that makes people indifferent to the choice between buying conventional rice ($E_{i0}$), and organic or traditional rice ($E_{i1}$), that is:

$$\alpha_0 \cdot (P_{i0} + WTP) + \alpha_{i1} \cdot E_{i1} + \beta_{i1} \cdot X_i = \alpha_0 \cdot P_{i0} + \alpha_{i0} \cdot E_{i0} + \beta_{i0} \cdot X_i \tag{2}$$

This results in $WTP = \frac{\alpha_{i0} - \alpha_{i1}}{\alpha_0} + \frac{\beta_{i0} - \beta_{i1}}{\alpha_0} X_i$, where $\frac{\beta_{i0} - \beta_{i1}}{\alpha_0}$ is the marginal effect of $X_i$ on consumer WTP for "modern" organic and sustainable traditional agricultural rice.

### 3.2. WTP Elicitation Method

In this paper, we use a payment card approach to measure consumer willingness to pay. Some research shows that WTP values estimated by a payment card approach are more robust than those relying on a dichotomous-choice approach [28], and there is no starting point bias [29]. The payment card approach is also prevalent in the current literature [18,27]. In the survey, we told the respondents that the average market price of conventional rice was 3 yuan/500 g (after a comprehensive market investigation, we found that the average market price of conventional rice was 3 yuan/500 g. Randall et al. found the starting point bias was not prominent in a payment card approach [30]). We then asked the respondents to pick one out of five listed prices that may best capture their WTP for "modern" organic and traditional rice. These listed prices were: "3.0–3.5 yuan", "3.5–4.0 yuan", "4.0–4.5 yuan", "5.0–6.0 yuan", and "6.0 yuan or above". The respondents needed to answer the following question:

> *Suppose that a bag of 500 g conventional rice is sold in-store at 3 yuan/500 g—which is the highest price that you are willing to pay for the XXX rice?*

One certified organic rice product and two typical traditional agricultural products are included in this study with the goal of helping us to compare consumers' preferences for "modern" organic and sustainable traditional agricultural products. Rice from the fish–rice system and rice fertilized by green manure are included in this study. Fish–rice systems and green manuring are the most popular and historically sustainable traditional farming methods in China. The integrated rice–fish system is an ancient sustainable practice employed by numerous farmers in China and some other countries in Asia. Fish are kept concurrently in a rice field to improve soil fertility [31]. Using green manure rather than synthetic fertilizers in agricultural production is also common and historical practice by traditional smallholder farmers, who provide food for themselves and their families or community.

*3.3. Econometric Model*

We used interval regression (we further examined the robustness of this finding by also conducting seemingly unrelated regression. Seemingly unrelated regression is usually more efficient than the models that estimate each equation independently [32]. The results of seemingly unrelated regression are consistent with interval regression and are outlined in Appendix A) to examine the important factors that affect respondents' WTP following Yang et al. [33]. In our payment card survey, each respondent chose a price range from the five listed options for "modern" organic or sustainable traditional agricultural products. Because each of the listed price ranges was interval-censored, previous studies show that interval regression is more efficient for this data than an ordered probit model [33,34].

Assuming a latent variable $WTP^*$ indicates an individual's true WTP, and the true WTP is assumed to lie in regions $(\alpha_1, \alpha_2], \ldots, (\alpha_J, +\infty)$, we can further assume that $WTP^*$ can be modeled as a linear function:

$$WTP^* = X_i'\beta + u_i \tag{3}$$

where $X_i$ is a vector of demographic and preference variables and $u_i$ is the random error following a normal distribution. An individual's selection of a specific price range will be then be determined by the following functions:

$$WTP = 1 \text{ if } \alpha_1 \leq WTP^* \leq \alpha_2 \tag{4}$$

$$WTP = 2 \text{ if } \alpha_2 \leq WTP^* \leq \alpha_3 \tag{5}$$

$$\ldots$$

$$WTP = J \text{ if } WTP^* > \alpha_J \tag{6}$$

where $\alpha_{j-1}$ and $\alpha_j$ defines the upper and lower boundary of the price range. The probability that a respondent chooses the range between upper and lower boundaries is then:

$$\Pr\left[\alpha_{j-1} < WTP^* \leq \alpha_j\right] = \Pr\left[WTP^* \leq \alpha_j\right] - Pr\left[WTP^* \leq \alpha_{j-1}\right] = F^*(\alpha_j) - F^*(\alpha_{j-1}) \tag{7}$$

F is a cumulative normal distribution. Maximum likelihood estimation can be used to obtain consistent estimates of the parameter vector $\beta$ and the error standard deviation $\sigma$.

## 4. Data and Empirical Results

*4.1. Data and Descriptive Statistics*

A consumer survey company (www.wjx.cn) was employed to distribute our survey online to its national representative consumer panels in China in September 2017. Several measures were taken to ensure the quality of the survey. Firstly, respondents were randomly selected from a 2.6 million sample library. Secondly, the company only allows one IP address, one computer, and one account for one questionnaire at the same time. Thirdly, some specified requirements were satisfied, including that respondents had to be 18 years old or older and were the household's primary shoppers. At last, we implemented the "trap question" method. We included both of the statements "I am an environmentalist" and "I am not an environmentalist" in the survey. If the answer is the same, we determine that the participant is less careful in answering survey questions. This method [35,36] was used to identify the respondents who may not carefully read the survey questions. In total, 2103 people responded to the survey. Removing the respondents who failed the "trap questions" and those with any missing responses resulted in 1422 observations for final analysis.

The survey included questions about consumers' demographic characteristics, perceptions, and WTP for "modern" organic or sustainable traditional rice. Table 2 reports

the sample demographic statistics. About 53.09% of the respondents were females because females shop for groceries such as agricultural products more frequently than males in China. More than 50% of respondents' household monthly income was under 12,000 RMB. About 87% of the respondents had a college or postgraduate degree, and 61% were aged 34 years old or younger. In addition, more than 75% of our samples had children at home. Our sample can be a good representative of the middle class in China, who are young, have relatively high incomes, and have the greatest purchasing power in the Chinese food market [37]. The rapidly growing demand for value-added agricultural products is mainly promoted by the growing middle class in China [38], and most of them have a bachelor's degree [39].

**Table 2.** Description of the sample: socio-demographic characteristics.

| Variables | Categories | Sample Size | Percent Sample (%) |
|---|---|---|---|
| Gender | Male | 667 | 46.91 |
| | Female | 755 | 53.09 |
| Age | Age (≤25) | 230 | 16.17 |
| | Age (26–34) | 642 | 45.15 |
| | Age (35–44) | 387 | 27.22 |
| | Age (>44) | 163 | 11.46 |
| Education | High school degree or less | 183 | 12.87 |
| | Bachelor's degree or Associate degree | 1135 | 79.82 |
| | Postgraduate degree (MS or doctoral) | 104 | 7.31 |
| Household Monthly Income (¥) | Income (<8000) | 345 | 24.26 |
| | Income (8001–12,000) | 408 | 28.69 |
| | Income (12,001–16,000) | 297 | 20.89 |
| | Income (16,001–20,000) | 210 | 14.77 |
| | Income (>20,001) | 162 | 11.39 |
| Has children | No | 349 | 24.54 |
| | Yes | 1073 | 75.46 |
| Sample Size | | 1422 | 100% |

*4.2. Perception of "Modern" Organic and Sustainable Traditional Agricultural Rice*

Figure 1 reports the statistics of respondents' perception of "modern" organic and sustainable traditional agricultural products. In this study, respondents were asked whether they perceived "modern" organic or sustainable traditional agricultural rice as safer/more environment-friendly than conventional rice. Figure 1 shows that most respondents considered "modern" organic and sustainable traditional agricultural rice as safer/more environment-friendly. Interestingly, there were more respondents who agreed or strongly agreed that traditional agricultural rice was safer/more environment-friendly than "modern" organic rice. The reason may be that consumers lack awareness and a good understanding of what the term "organic" means [40]. Hence, consumers have heterogeneous interpretations of what "organic" is [14], and consumers' interest in such labeling cannot be taken for granted [41]. Consumers can still perceive the benefit of sustainable traditional agriculture without labels because domestic consumers are familiar with sustainable traditional agricultural techniques. Therefore, they can directly infer the information about the quality of food from sustainable traditional agriculture.

Table 3 reports the statistics of respondents' WTP premium. We compute the means of WTP premium by assuming the true value is the middle point of the interval (the WTP premium value for the up-open interval is equal to the sum of the lower boundary and the half distance of the neighboring interval). The mean WTP premium values of rice from the fish–rice system, rice fertilized by green manure and "modern" organic rice are 1.23 yuan, 1.11 yuan, and 0.791 yuan, respectively, for the whole sample. The differences between the WTP for rice from the fish–rice system, rice fertilized by green manure, and "modern" organic rice are all statistically significantly different from zeros (Table 3).

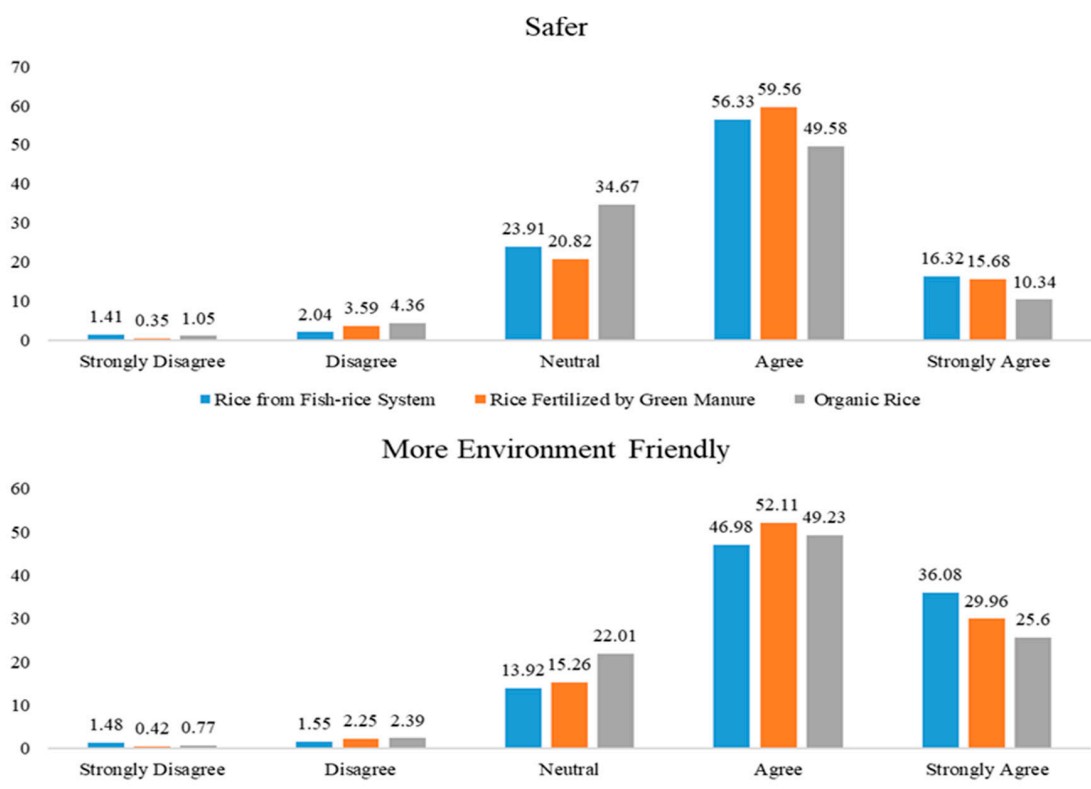

**Figure 1.** Respondents' perception of "modern" organic and sustainable traditional agricultural products.

**Table 3.** Description of WTP premium by socio-demographic characteristics.

| Variables | Categories | Rice from Fish–Rice System | | Rice Fertilized by Green Manure | | "Modern" Organic Rice | |
|---|---|---|---|---|---|---|---|
| | | Mean (yuan) | Std. Dev | Mean (yuan) | Std. Dev | Mean (yuan) | Std. Dev |
| Gender | Male | 1.182 | 0.740 | 1.079 | 0.740 | 1.136 | 0.759 |
| | Female | 1.266 | 0.783 | 1.134 | 0.783 | 1.219 | 0.817 |
| Age | Age (≤25) | 1.037 | 0.753 | 1.000 | 0.753 | 0.983 | 0.677 |
| | Age (26–34) | 1.263 | 0.729 | 1.122 | 0.729 | 1.223 | 0.780 |
| | Age (35–44) | 1.271 | 0.781 | 1.151 | 0.781 | 1.223 | 0.780 |
| | Age (>44) | 1.244 | 0.837 | 1.103 | 0.837 | 1.221 | 0.900 |
| Education | High school degree or less | 0.988 | 0.770 | 0.919 | 0.770 | 0.919 | 0.668 |
| | Bachelor's degree or Associate degree | 1.266 | 0.756 | 1.133 | 0.756 | 1.222 | 0.795 |
| | Postgraduate degree (MS or doctoral) | 1.214 | 0.776 | 1.166 | 0.776 | 1.183 | 0.860 |
| Family Income (RMB/month) | Income1 (<8000) | 0.967 | 0.694 | 0.893 | 0.694 | 0.907 | 0.673 |
| | Income2 (8001–12,000) | 1.142 | 0.690 | 1.026 | 0.690 | 1.099 | 0.729 |
| | Income3 (12,001–16,000) | 1.253 | 0.726 | 1.124 | 0.726 | 1.239 | 0.776 |
| | Income4 (16,001–20,000) | 1.451 | 0.724 | 1.293 | 0.724 | 1.419 | 0.784 |
| | Income5 (>20,001) | 1.650 | 0.928 | 1.505 | 0.928 | 1.549 | 0.954 |
| Has children | Do not have children | 1.048 | 0.736 | 0.943 | 0.736 | 0.968 | 0.684 |
| | Have children | 1.284 | 0.764 | 1.162 | 0.764 | 1.249 | 0.811 |
| Whole sample | Whole sample | 1.226 | 0.764 | 1.108 | 0.670 | 1.180 | 0.791 |

Note: We find a statistically significant difference in the mean WTP among these agricultural products: *t*-stat = 4.57 *** (*t* test for "modern" organic rice and rice fertilized by green manure), *t*-stat = 3.48 *** (*t* test for "modern" organic rice and rice from Fish–rice System), *t*-stat = 8.97 *** (*t* test for rice from fish–rice system and rice fertilized by green manure).

Female respondents and those aged 35–44, with high family income, or having children are willing to pay higher prices for both sustainable traditional agricultural products and "modern" organic products. These results imply that sustainable traditional agricultural

products may be an advisable substitution for expensive "modern" organic products in developing countries. This is especially true when there are serious food safety issues, and most families cannot afford the high prices of "modern" organic products.

### 4.3. Factors Affecting Consumer WTP

We estimated both the interval regression (Table 4) and seemingly unrelated regression (Table A1) using the individual-level WTPs as dependent variables and perception and demographics as explanatory variables. The regression results in both models are quite consistent regarding the coefficient signs and significant levels, and the results from both models are consistent (*p*-value = 0.000). The dependent variables in the interval regression were the right- and left-censored WTP for rice from the different kinds of rice, and the dependent variables in the seemingly unrelated regression were the means of WTP. Demographic variables (such as age, gender, income, education, and whether the respondents had children) and perception variables [26] were included in the independent variables.

**Table 4.** Result of the interval regression model.

| Parameter | Interval Regression Model | | |
|---|---|---|---|
| | Rice from Fish–rice System (1) | Rice Fertilized by Green Manure (2) | Organic Rice (3) |
| Safer | 0.0956 *** | 0.0441 | 0.122 *** |
| | (0.0321) | (0.0278) | (0.0298) |
| More Environment-Friendly | 0.0128 | 0.0486 * | 0.0568 ** |
| | (0.0301) | (0.0268) | (0.0286) |
| Female | 0.0668 * | 0.0315 | 0.0542 |
| | (0.0370) | (0.0340) | (0.0383) |
| Age | 0.00302 | 0.00129 | 0.00402 * |
| | (0.0021) | (0.0020) | (0.0022) |
| Income (2) | 0.132 ** | 0.104 ** | 0.125 ** |
| | (0.0520) | (0.0478) | (0.0538) |
| Income (3) | 0.216 *** | 0.182 *** | 0.231 *** |
| | (0.0568) | (0.0522) | (0.0589) |
| Income (4) | 0.388 *** | 0.325 *** | 0.369 *** |
| | (0.0632) | (0.0581) | (0.0656) |
| Income (5) | 0.585 *** | 0.523 *** | 0.501 *** |
| | (0.0687) | (0.0632) | (0.0713) |
| Bachelor's degree | 0.133 ** | 0.0804 | 0.141 ** |
| | (0.0582) | (0.0535) | (0.0600) |
| Postgraduate degree | 0.0316 | 0.0531 | 0.0506 |
| | (0.0883) | (0.0812) | (0.0912) |
| Has children | 0.154 *** | 0.154 *** | 0.182 *** |
| | (0.0430) | (0.0396) | (0.0446) |
| Intercept | 3.151 *** | 3.273 *** | 2.828 *** |
| | (0.1420) | (0.1350) | (0.1480) |

Note: * Indicates statistically significant at the 10% significance level; ** indicates statistically significant at the 5% significance level; *** indicates statistically significant at the 1% significance level.

Results from the interval regression show that consumers' perception of food safety of rice from the fish–rice system was significantly positive at the 1% significance level. Consumers' perception of the environmental characteristics of rice fertilized by green manure was significantly positive at the 10% significance level. For "modern" organic rice, both perception variables were significantly positive at the 10% and 5% significance levels, respectively. The reason for such results is that the key characteristics of these agricultural techniques are quite different. The most important characteristic of rice from the fish–rice system is food safety because it uses fewer pesticides. In contrast, the key advantage of green manures is soil improvement and soil protection, which can play an important role in sustainable cropping systems. Furthermore, organic farming usually does not use synthetic pesticides and fertilizers and genetically modified organisms. Therefore, both food safety

and environmental perceptions of "modern" organic rice affect consumer WTP. The result implies that in addition to certificated organic products, consumers are also willing to pay higher prices for sustainable traditional agricultural products, and consumers are only willing to pay for the key characteristics of these farming methods. Therefore, promoting the key feature of sustainable traditional agricultural techniques may be more effective in increasing the demand for sustainable agricultural products. Improving consumers' knowledge about these sustainable traditional agricultural techniques can also increase consumers' premiums.

Having children at home had significant effects on consumer WTP for both "modern" organic and sustainable traditional agricultural rice. All household income levels have significant effects on consumers' WTP for both "modern" organic and traditional agricultural rice. Compared to the respondents with a high school degree or less, those with bachelor's or associate degrees might be willing to pay higher prices for organic rice and rice from the rice–fish system (at the 10% significance level). Females might be willing to pay higher prices for the rice–fish system rice and elderly consumers are more willing to pay for "modern" organic food. Spending on healthy food has seen tremendous growth over the last decade in China, especially for female, elderly, and high-income consumers. These consumers have an increasing interest in "partly natural" or organic food and are willing to pay a premium for healthy food, providing a huge food market for "modern" organic food and sustainable traditional agriculture in developing countries.

### 4.4. Discussion

The study results show that the mean WTP premium values are close to Gao et al.'s study, which finds that the premiums for sustainable milk are about 40% [26]. This indicates that consumers are more willing to pay for sustainable traditional agricultural products than "modern" organic products. It is consistent with Kim et al.'s study, which shows at least 33% of consumers would not make an extra payment for organic products because they think organic products are inferior in taste, safety or appearance [13]. In addition, Ortega et al. also conclude that Chinese consumers prefer green food to organic certification [42]. This is because organic certification is poorly understood in the Chinese food market [43,44].

Furthermore, this study suggests potential inter-generational concern for sustainable development, which is consistent with Gao et al.'s study [26]. It is also consistent with some previous research that the consumption of organic or green food is significantly influenced by household income [27,45]. This result implies that there is great demand potential for both "modern" organic and sustainable traditional agricultural products in China, where the Gross Domestic Product (GDP) per capita has continued to increase in past decades.

## 5. Implications and Conclusions

For centuries, many smallholder farmers in developing countries have used sustainable traditional agricultural techniques to ensure food safety and sustainability. However, the value of sustainable traditional agricultural products, especially as an inexpensive substitution for expensive "modern" organic products in developing countries, is rarely studied. Taking two kinds of sustainable traditional agricultural products (rice from the fish–rice system and rice fertilized by green manure) as examples, this paper identifies and compares Chinese consumer preferences for sustainable traditional agricultural products and "modern" organic products and how they are willing to pay for each. The results of this study would provide vital information to identify the market space for millions of smallholder farmers to revitalize sustainable traditional agriculture and sustain farming practices that are more eco and environment-friendly than conventional modern agriculture.

As our results show, the average WTP values of the two kinds of sustainable traditional agricultural products are higher than those of "modern" organic products. The mean WTP values (over that of conventional rice) for rice from the fish–rice system, rice fertilized by green manure and "modern" organic rice are 1.226 yuan, 1.108 yuan and 0.791 yuan, respectively. The results from the interval regression and seemingly unrelated regression

further show that the perception of the key characteristics of sustainable traditional agricultural techniques has significantly affected consumers' willingness to pay. In contrast, both food safety and environmental perceptions affect consumers' willingness to pay for "modern" organic products. These findings imply that sustainable traditional agriculture can signal environment-friendly practices or food safety to consumers. The development of sustainable traditional agriculture could be an efficient and inexpensive organic movement in developing countries.

The organic movement is far from being homogeneous [46]. Considering the increasing demand for sustainable traditional agricultural products, revitalizing sustainable traditional agriculture may be a good way for developing countries to balance sustainability and economic feasibility. The key features of sustainable traditional farming techniques could be promoted to increase consumer demand. Helping millions of smallholder farms to promote sustainable traditional agricultural products could be one of the low-cost ways to achieve sustainable development in developing countries.

This paper provides a new perspective for balancing the sustainability and feasibility of agriculture in developing countries. However, several important questions are still worth studying. Firstly, as this paper only takes rice as an example, consumers' preferences for other sustainable traditional agricultural and "modern" organic products, such as vegetables and meat, should be studied. Secondly, one of the limitations of this study is that our sample includes more educated and young people, which is the disadvantage of an online survey. Future studies can recruit survey participants from more traditional shopping outlets who are usually less educated and have lower incomes than those participating in the online survey. Thirdly, since the survey was conducted in 2017, future studies could be done to identify whether significant differences raise in consumer perception and preferences toward "modern" organic and sustainable traditional agricultural products with the impact of online social media and the shock of the COVID-19 pandemic [47].

**Funding:** This research received no external funding.

**Institutional Review Board Statement:** Not applicable.

**Informed Consent Statement:** Informed consent was obtained from all subjects involved in the study.

**Data Availability Statement:** The data presented in this study are available on request from the corresponding author. The data are not publicly available due to privacy restrictions.

**Conflicts of Interest:** The authors declare no conflict of interest.

## Appendix A

**Table A1.** Regression results of SUR model.

| Parameter | Rice from Fish–Rice System (1) | Rice Fertilized by Green Manure (2) | Organic Rice (3) |
|---|---|---|---|
| Safer | 0.0921 *** | 0.0207 | 0.0951 *** |
| | (0.0215) | (0.0182) | (0.0204) |
| More Environment-Friendly | 0.0079 | 0.0158 | 0.0070 |
| | (0.0203) | (0.0177) | (0.0199) |
| Female | 0.0726 * | 0.0406 | 0.0653 |
| | (0.0387) | (0.0357) | (0.0398) |
| Age | 0.0034 | 0.0021 | 0.0047 ** |
| | (0.0022) | (0.0020) | (0.0023) |
| Income (2) | 0.1358 ** | 0.1078 ** | 0.1410 *** |
| | (0.0545) | (0.0502) | (0.0561) |
| Income (3) | 0.2246 *** | 0.1893 *** | 0.2555 *** |
| | (0.0595) | (0.0548) | (0.0612) |
| Income (4) | 0.4021 *** | 0.3427 *** | 0.4069 *** |
| | (0.0659) | (0.0608) | (0.0679) |

**Table A1.** *Cont.*

| Parameter | Rice from Fish–Rice System (1) | Rice Fertilized by Green Manure (2) | Organic Rice (3) |
|---|---|---|---|
| Income (5) | 0.6114 *** | 0.5546 *** | 0.5415 *** |
| | (0.0716) | (0.0661) | (0.0738) |
| Bachelor's degree | 0.1349 ** | 0.0900 | 0.1536 ** |
| | (0.0609) | (0.0562) | (0.0626) |
| Postgraduate degree | 0.0301 | 0.0642 | 0.0590 |
| | (0.0924) | (0.0853) | (0.0950) |
| Has children | 0.1600 *** | 0.1632 *** | 0.2059 *** |
| | (0.0450) | (0.0415) | (0.0463) |
| Intercept | 3.1707 *** | 3.4461 *** | 3.0529 *** |
| | (0.1331) | (0.1242) | (0.1378) |
| Correlation of the residuals | $\rho_{12}$= 0.736 | $\rho_{13}$= 0.697 | $\rho_{23}$= 0.715 |
| | | chi$^2$(3) = 2187.809, $p$ = 0.0000 | |

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
