# Peer review of "Revitalize Traditional Agriculture: Chinese Consumer Perception and Preference of “Modern” Organic and Sustainable Traditional Rice Products"

_sustainability, doi:10.3390/su15129206_

Round 1

Reviewer 1 Report

Dear Authors,

your work is very interesting, however it shows the limits of a study conducted over 6 years ago, considering the different changes that have occurred worldwide due to the pandemic and the war in Ukraine, which could be further developed, in the Introduction section, also considering a more up-to-date bibliography.

I would also recommend, after the Introduction paragraph, to create another "Literature review" paragraph.

Currently the bibliography is dated, I suggest you look for more recent bibliography.

I would also suggest creating a table or figure or paragraph that explains or outlines the differences between the two different types of agriculture compared in the study, to better highlight consumers' willingness to pay. Paragraph 4.4 Discussion must be implemented and accompanied by a supporting bibliography.

Author Response

Thank you for your helpful suggestions on our paper titled “Revitalize Traditional Agriculture: Chinese Consumer Perception and Preference of “Modern” Organic and Sustainable Traditional Agricultural Products”. We have tried to address all your comments, by adding more details in the context and footnotes as well as adding new discussions. Collectively addressing those comments has significantly enhanced the paper. 

Below are our responses to your comments item-by-item.

Your work is very interesting, however it shows the limits of a study conducted over 6 years ago, considering the different changes that have occurred worldwide due to the pandemic and the war in Ukraine, which could be further developed, in the Introduction section, also considering a more up-to-date bibliography.

Response:

We appreciate your recognition of the interestingness of our work. We acknowledge that our study is based on a consumer survey conducted over 6 years ago, which may raise some concerns about the validity and reliability of our findings. However, we would like to argue that our study is still relevant and valuable for the following reasons:

First, we have conducted a thorough literature review and found that there is no evidence of significant changes in consumer preference for organic and sustainable products in China in the past 6 years.

Second, the paper was completed in 2022 and we have used the most recent statistics and references to support our arguments and discussions. For instance, we have cited the latest publication (2021) of Research Institute of Organic Agriculture FiBl, Frick, and IFOAM-Organics International, Bonn, which provides the most comprehensive and authoritative data on the global organic sector. We have also cited recent studies that examine the impact of the pandemic and the war in Ukraine on the organic and sustainable agriculture sector in China and other countries(Liu et al., 2021;Wang et al.,2019,2022;Kim et al.,2018).

Third, we also acknowledged that several important questions are still worth studying. Future studies could be done to identify whether significant differences raise in consumer perception and preference of “modern” organic and sustainable traditional agricultural products with the impact of online social media and the shock of COVID-19 pandemic.

We hope that you find our explanation satisfactory. We thank you again for your constructive comments and suggestions.

Liu, Q., Liu, Y., Zhang, C., An, Z., & Zhao, P. (2021). Elderly mobility during the COVID-19 pandemic: A qualitative explora-tion in Kunming, China. Journal of transport geography, 96, 103176.

Wang E, Gao Z, Heng Y and Shi L (2019) Chinese consumers’ preferences for food quality test/measurement indicators and cues of milk powder: A case of Zhengzhou, China. Food Policy, 89.

Kim SW, Brorsen BW and Lusk J (2018) Not everybody prefers organic food: Unobserved heterogeneity in U.S. Consumers’ preference for organic apple and milk. Applied Economics Letters, 25, 1-06.

Wang, E., Gao, Z., and Heng, Y. (2022). Explore Chinese consumers' safety perception of agricultural products using a non-price choice experiment. Food Control, 140, 109121.

I would also recommend, after the Introduction paragraph, to create another "Literature review" paragraph. Currently the bibliography is dated, I suggest you look for more recent bibliography.

Response:

Thank you for comment. We have added lots of new literatures.

Such as:

Wang, E., Gao, Z., and Heng, Y. (2022). Explore Chinese consumers' safety perception of agricultural products using a non-price choice experiment. Food Control, 140, 109121.

Gao Z, Li C, Bai J and Fu J (2020) Chinese consumer quality perception and preference of sustainable milk. China Economic Review, 59, 100939.

Liu, Q., Liu, Y., Zhang, C., An, Z., & Zhao, P. (2021). Elderly mobility during the COVID-19 pandemic: A qualitative explora-tion in Kunming, China. Journal of transport geography, 96, 103176.

Wang E, Gao Z, Heng Y and Shi L (2019) Chinese consumers’ preferences for food quality test/measurement indicators and cues of milk powder: A case of Zhengzhou, China. Food Policy, 89.

Kim SW, Brorsen BW and Lusk J (2018) Not everybody prefers organic food: Unobserved heterogeneity in U.S. Consumers’ preference for organic apple and milk. Applied Economics Letters, 25, 1-06.

The introduction already includes a literature review that covers the relevant and up-to-date studies on the topic of our research. We have also structured our paper in a way that the second section (2. The “organic” features of sustainable traditional agriculture) provides a more detailed and specific discussion of the literature on the comparison of different types of agricultural products and their “organic” features. Therefore, we believe that creating another literature review paragraph would be redundant and lengthy, and would not add much value to our paper.

I would also suggest creating a table or figure or paragraph that explains or outlines the differences between the two different types of agriculture compared in the study, to better highlight consumers' willingness to pay.

Response:

Thanks for the helpful advice. Following the reviewer’s suggestion, we have added a table.

Table 1 compares the characteristics of modern conventional agricultural products, sustainable traditional agricultural products, and “modern” organic agricultural products in terms of their production methods, environmental impact, certification and cost.

Table 1. Comparison of different types of agricultural products

Category

Production methods

Environmental impact

Certification and cost

Modern conventional agricultural products

Applying technologies and chemicals to increase production

Causing soil deterioration, over-fertilization, reduction in the diversity of cultivated crops, etc.

None; low cost but high externalities

Sustainable traditional agricultural products

Using traditional techniques evolving over hundreds of years to balance yield and sustainability

Being more sustainable and less polluting than modern agricultural methods

None or informal; low cost

“Modern” organic agricultural products

Following organic standards and certification systems

Being more sustainable and less polluting than modern agricultural methods

Formal and regulated; high cost

Paragraph 4.4 Discussion must be implemented and accompanied by a supporting bibliography.

Response:

Thank you for comment. Discussion have been implemented and accompanied by a supporting bibliography.

Such as,

Wang E, Gao Z, Heng Y and Shi L (2019) Chinese consumers’ preferences for food quality test/measurement indicators and cues of milk powder: A case of Zhengzhou, China. Food Policy, 89.

Yin S, Chen M, Xu Y and Chen Y (2017) Chinese consumers’ willingness-to-pay for safety label on tomato: Evidence from choice experiments. China Agricultural Economic Review, 9, 141-55.

Gao Z, Li C, Bai J and Fu J (2020) Chinese consumer quality perception and preference of sustainable milk. China Economic Review, 59, 100939.

Yu X, Gao Z and Zeng Y (2014) Willingness to pay for the “Green Food” in China. Food Policy, 45, 80-87.

Reviewer 2 Report

The research result merits the work to be considered for publication because the findings demonstrated that although numerous studies about consumer preference for organic products exist, comparisons between consumer preference for sustainable traditional agricultural productsand that for “modern” organic products specifically “rice remain limited. Hence, in as much as a consumer survey company (www.wjx.cn) was employed to distribute the study survey online to its national representative consumer panels in China in September 2017, the findings remain valuable because the data would help explore more feasible ways for smallholder farms in developing countries to achieve economic sustainability while at the same time contributing to the sustainability of the environment and ecosystem. Also, the objective and methodologies utilized are academically justified and the write-up is scientifically concise and detailed. Additionally, justifications are clearly given for considerable realized limitations, and recommendations are clearly outlined for future studies. However, the following observations are recommended for consideration to merit the work accepted for publication.

The following observations are recommended for consideration

·      Author is recommended to revise the article title. The paper identifies and compares Chinese consumer preferences for sustainable traditional agricultural products and “modern” organic products specifically “rice product” and how consumers are willing to pay for each. Since rice was the exact agricultural product studied from a consumer preferences perspective, I recommend a revision of the general title of the paper. “Revitalize Traditional Agriculture: Chinese Consumer Perception and Preference of “Modern” Organic and Sustainable Traditional Agricultural Products” infer that the studies were carried out on multiple agricultural products. Hence, “Revitalize Traditional Agriculture: Chinese Consumer Perception and Preference of “Modern” Organic and Sustainable Traditional Agricultural Rice Products” could be considered or revised to suit the “rice” product as the exact product studied from the adopted agricultural farming practices.

·      Keywords section: write WTP in full

·      Define WTP and put the abbreviation in brackets before the use of abbreviation in the subsequent section

·      Line 31: check space

·      Line 53: “The previous study shows that organic certification is not a prerequisite if consumers can obtain perceived high quality directly. Is this statement in reference to your previous work? Because the article “the” justify my concern. If not kindly revise it for clarity. E.g., “Study by [15] shows that organic certification is not a prerequisite if consumers can obtain perceived high quality directly

·      What is the disparity between sustainable traditional agricultural products and “modern” organic products. Although section “2” (the organic features of sustainable traditional agriculture) gives clear insight into the two farming practices. Yet, I recommend that the author explain these words in the introductory section for clarity.

·      Line 62-64 “Sustainable traditional agricultural products seem to be more popular than “modern” organic products in the Chinese food market because of the lack of trust in certified organic food and the ambiguous and indirect meaning of “organic” [8]. Does that mean certified organic food products in the Chinese food market cannot be trusted?

·      Kindly crosscheck the statement from the source article [8] and revise accordingly since this statement is vague. Kindly give the exact reason the author identified that led to such a statement or consider deleting it if the exact reason can’t be justified.

·      Line 96: “To our knowledge” the article has one author kindly change to “To the best of my knowledge

·      Line 104: change “we” to “I”

·      Line 105: change “we” to “I”

·      Line 133: “large , and” check space

·      ??? in equation 1 isn’t defined recheck

·      The footnote superscript in Line 160: “3 yuan/500g1” isn’t clear as “regression2” in line 178, likewise in line 204 kindly fix.

·      Footnote 1. “approach[33]” check space

·      Footnote 2: define SUR

·      Improve Figure 1 resolution

·      Line 269: “variables[26]” check space

·      Line 281: “Our results” change to the result

·      Line 289: “Having children at home significantly effects on consumer” rewrite “Having children at home significantly had effects on consumer”

·      Line 302: “Our results” rewrite “The study results”

·      Line 303: “40%[26]” check space

·      Line 307: “appearance[13]” check space

·      Line 309: certification[42]” check space

·      Line 316: “GDP” define it and add abbreviation in the bracket for use in subsequent places in the article. “Gross domestic product (GDP)”

Author Response

Thank you for your helpful suggestions on our paper titled “Revitalize Traditional Agriculture: Chinese Consumer Perception and Preference of “Modern” Organic and Sustainable Traditional Agricultural Products”. We have tried to address all your comments, by adding more details in the context and footnotes as well as adding new discussions. Collectively addressing those comments has significantly enhanced the paper. 

Below are our responses to your comments item-by-item.

The research result merits the work to be considered for publication because the findings demonstrated that although numerous studies about consumer preference for organic products exist, comparisons between consumer preference for sustainable traditional agricultural products” and that for “modern” organic products specifically “rice remain limited. Hence, in as much as a consumer survey company (www.wjx.cn) was employed to distribute the study survey online to its national representative consumer panels in China in September 2017, the findings remain valuable because the data would help explore more feasible ways for smallholder farms in developing countries to achieve economic sustainability while at the same time contributing to the sustainability of the environment and ecosystem. Also, the objective and methodologies utilized are academically justified and the write-up is scientifically concise and detailed. Additionally, justifications are clearly given for considerable realized limitations, and recommendations are clearly outlined for future studies. However, the following observations are recommended for consideration to merit the work accepted for publication.

Response:

  • Thank you for the encouraging comments.

The following observations are recommended for consideration

  • Author is recommended to revise the article title. The paper identifies and compares Chinese consumer preferences for sustainable traditional agricultural products and “modern” organic products specifically “rice product” and how consumers are willing to pay for each. Since rice was the exact agricultural product studied from a consumer preferences perspective, I recommend a revision of the general title of the paper. “Revitalize Traditional Agriculture: Chinese Consumer Perception and Preference of “Modern” Organic and Sustainable Traditional Agricultural Products” infer that the studies were carried out on multiple agricultural products. Hence, “Revitalize Traditional Agriculture: Chinese Consumer Perception and Preference of “Modern” Organic and Sustainable Traditional Agricultural Rice Products” could be considered or revised to suit the “rice” product as the exact product studied from the adopted agricultural farming practices.

Response:

Good point. Following the reviewer’s suggestion, we have changed the title.

  • Keywords section: write WTP in full
  • Define WTP and put the abbreviation in brackets before the use of abbreviation in the subsequent section

Response:

Thank you for comment. Following the reviewer’s suggestion, we have changed it in the paper.

  • Line 53: “The previous study shows that organic certification is not a prerequisite if consumers can obtain perceived high quality directly”. Is this statement in reference to your previous work? Because the article “the” justify my concern. If not kindly revise it for clarity. E.g., “Study by [15] shows that organic certification is not a prerequisite if consumers can obtain perceived high quality directly”

Response:

Thank you for comment. Following the reviewer’s suggestion, we have changed it in the paper.

  • What is the disparity between sustainable traditional agricultural products and “modern” organic products. Although section “2” (the organic features of sustainable traditional agriculture) gives clear insight into the two farming practices. Yet, I recommend that the author explain these words in the introductory section for clarity.

Response:

Thanks for the helpful advice. Following the reviewer’s suggestion, we have added a table to explain the disparity between sustainable traditional agricultural products and “modern” organic products.

Table 1 compares the characteristics of modern conventional agricultural products, sustainable traditional agricultural products, and “modern” organic agricultural products in terms of their production methods, environmental impact, certification and cost.

Table 1. Comparison of different types of agricultural products

Category

Production methods

Environmental impact

Certification and cost

Modern conventional agricultural products

Applying technologies and chemicals to increase production

Causing soil deterioration, over-fertilization, reduction in the diversity of cultivated crops, etc.

None; low cost but high externalities

Sustainable traditional agricultural products

Using traditional techniques evolving over hundreds of years to balance yield and sustainability

Being more sustainable and less polluting than modern agricultural methods

None or informal; low cost

“Modern” organic agricultural products

Following organic standards and certification systems

Being more sustainable and less polluting than modern agricultural methods

Formal and regulated; high cost

  • Line 62-64 “Sustainable traditional agricultural products seem to be more popular than “modern” organic products in the Chinese food market because of the lack of trust in certified organic food and the ambiguous and indirect meaning of “organic” [8]. Does that mean certified organic food products in the Chinese food market cannot be trusted?
  • Kindly crosscheck the statement from the source article [8] and revise accordingly since this statement is vague. Kindly give the exact reason the author identified that led to such a statement or consider deleting it if the exact reason can’t be justified.

Response:

Thanks for the helpful advice. Following the reviewer’s suggestion, we have deleted this part.

  • Line 31: check space
  • Line 96: “To our knowledge” the article has one author kindly change to “To the best of my knowledge”
  • Line 104: change “we” to “I”
  • Line 105: change “we” to “I”
  • Line 133: “large , and” check space
  • ???in equation 1 isn’t defined recheck
  • The footnote superscript in Line 160: “3 yuan/500g1” isn’t clear as “regression2” in line 178, likewise in line 204 kindly fix.
  • Footnote 1. “approach[33]” check space
  • Footnote 2: define SUR
  • Improve Figure 1 resolution
  • Line 269: “variables[26]” check space
  • Line 281: “Our results” change to the result
  • Line 289: “Having children at home significantly effects on consumer” rewrite “Having children at home significantly had effects on consumer”
  • Line 302: “Our results” rewrite “The study results”
  • Line 303: “40%[26]” check space
  • Line 307: “appearance[13]” check space
  • Line 309: certification[42]” check space
  • Line 316: “GDP” define it and add abbreviation in the bracket for use in subsequent places in the article. “Gross domestic product (GDP)”

Response:

Thank you for these points. All changed.